# Healthcare consumers' perceptions of incentive-linked prescribing: A scoping review

**Muhammad Naveed Noor** [1,2], **Haider Safdar Abbasi**[2], **Nina van Der Mark**[3], **Zahida Azizullah**[2], **Janice Linton** [4], **Afifah Rahman-Shepherd** [3], **Amna Rehana Siddiqui** [5], **Mishal Sameer Khan** [3]*, **Rumina Hasan**[2], **Sadia Shakoor** [2]

1 Department of Community Health Sciences, Institute for Global Public Health, University of Manitoba, Winnipeg, Manitoba, Canada, 2 Department of Pathology and Laboratory Medicine, Aga Khan University, Karachi, Pakistan, 3 Department of Global Health and Development, London School of Hygiene and Tropical Medicine, London, United Kingdom, 4 Neil John Maclean Health Sciences Library, University of Manitoba, Winnipeg, Manitoba, Canada, 5 Jinnah Sindh Medical University, Karachi, Pakistan

* mishal.khan@lshtm.ac.uk

**Data Availability Statement:** All relevant data are within the paper and its Supporting Information files.

## Abstract

Incentive-linked prescribing (ILP) is considered a controversial practice universally. If incentivised, physicians may prioritise meeting pharmaceutical sales targets through prescriptions, rather than considering patients' health and wellbeing. Despite the potential harms of ILP to patients and important stakeholders in the healthcare system, healthcare consumers (HCCs) which include patients and the general public often have far less awareness about the practice of pharmaceutical incentivisation of physicians. We conducted a scoping review to explore what existing research says about HCCs' perceptions of the financial relationship between physicians and pharmaceutical companies. To conduct this scoping review, we followed Arksey and O'Malley's five-stage framework: identifying research questions, identifying relevant studies, selecting eligible studies, data charting, and collating, summarising, and reporting results. We also used the Preferred Reporting Items for Systematic Reviews and Meta-Analyses' extension for scoping reviews (PRISMA-ScR), as a guide to organise the information in this review. Quantitative and qualitative studies with patients and the general public, published in the English language were identified through searches of Scopus, Medline (OVID), EMBASE (OVID), and Google Scholar. Three themes emerged through the analysis of the 13 eligible studies: understanding of incentivisation, perceptions of hazards linked to ILP, and HCCs' suggestions to address it. We found documentation that HCCs exhibited a range of knowledge from good to insufficient about the pharmaceutical incentivisation of physicians. HCCs perceived several hazards linked to ILP such as a lack of trust in physicians and the healthcare system, the prescribing of unnecessary medications, and the negative effect on physicians' reputations in society. In addition to strong regulatory controls, it is critical that physicians self-regulate their behaviour, and publicly disclose if they have any financial ties with pharmaceutical companies. Doing so can contribute to trust between patients and physicians, an important part of patient-focused care and a contributor to user confidence in the wider health system.

**Funding:** This work is supported by the UK Research and Innovation (UKRI) funding agency (MR/T02349X/1 to MSK). The funder had no role in study design, data collection and analysis, decision to publish, or preparation of the manuscript.

**Competing interests:** I have read the journal's policy and two authors of this manuscript have the following competing interests: Dr Mishal Khan and Dr Sadia Shakoor are part of the editorial board of PLOS Global Public Health. All other others have no competing interests to declare.

## Introduction

Unethical drug promotion is a widely used practice by pharmaceutical companies, in which sales representatives offer incentives to physicians in exchange for prescriptions of their companies' drugs. This practice is thought to distort physicians' prescribing behaviour [1–3]. When incentivised, physicians may choose to prescribe drugs, which are either unnecessary or expensive [4]. Consequently, incentive-linked prescribing (ILP) may negatively affect patients' health or exacerbate their financial difficulty [4–6].

While there is a wealth of studies on ILP, most of them are conducted with physicians, the pharmaceutical industry, and other relevant stakeholders (i.e., regulators or nongovernmental organisations) [5–8]. This research is important and adds to our understanding of the mechanisms through which ILP happens and the factors that mediate this process. Because patients are also important stakeholders in the healthcare system and perhaps the most at risk of suffering the financial or physical consequences of ILP, it is critical to take their perspectives into account. In the past 20 years, many researchers from various parts of the world attempted to explore the perceptions of patients and the general public regarding the financial relationship between physicians and the pharmaceutical industry. Both population groups may be better described as healthcare consumers (HCCs), due to the experience of their consumption of healthcare services as well as pharmaceutical products.

Trust in physicians is considered a key strength of any healthcare system [9]. In settings, where HCCs lack trust in physicians, a decreased adherence to the treatment recommended by physicians is noted [10]. There is strong evidence that ILP happens at a large scale specifically in low- and middle-income countries (LMICs) [4,11–13]. Nevertheless, little is known about what views HCCs hold regarding the financial relationship between physicians and pharmaceutical companies, the extent to which it can affect their trust in physicians, and what they think about how incentivization of physicians by pharmaceutical companies may affect them. Understanding HCCs' views is critical for informing policies and practices around ethical pharmaceutical marketing and prescribing, towards sustaining patients' trust in healthcare professionals and the healthcare sector more broadly.

The most recent systematic review in this field was conducted in 2016, with a focus on the knowledge, beliefs and attitudes of HCCs around interactions of physicians with pharmaceutical and device industries offering a quantitative analysis of the acceptability of physicians' financial engagement with these industries [14]. Our scoping review provides a detailed narrative analysis of HCCs' perspectives of pharmaceutical incentivisation to physicians and significantly complements previous knowledge synthesis in the field. Our analysis specifically offers several important messages for physicians and other health professionals worldwide, orienting their attention to the level of awareness amongst HCCs about unethical exchanges taking place in clinical settings, and how this may affect physician reputation in society. Since little is known about the level of HCCs' awareness of ILP and how this practice is perceived by and has affected them, a scoping review of the existing research is essential to inform policy and practice globally, regionally and in individual clinical settings.

## Research objective and questions

The overarching objective of the review is to present an analysis of the published evidence on HCC's perspectives regarding pharmaceutical incentivisation to physicians. Our main question was what does existing research say about HCCs' perceptions of the relationship between physicians and pharmaceutical companies? The specific questions that we aimed to answer through this review included:

1. How aware are HCCs about ILP and what information do they have regarding the incentive types given to physicians?

2. What are HCCs' perceptions of the effects of pharmaceutical incentivisation of physicians in exchange for prescriptions?

3. What actions do HCCs think are necessary to address the pharmaceutical incentivization of physicians?

## Materials and methods

A protocol to conduct this scoping review was registered with Open Science Framework [15]. We used a five-stage framework developed by Arksey and O'Malley. These stages include identifying research questions, identifying relevant studies, selecting eligible studies, data charting, and collating, summarising, and reporting results [16]. To present information in this article, we employed the Preferred Reporting Items for Systematic Reviews and Meta-Analyses' extension for scoping reviews (PRISMA-ScR), as a guide [17]. The inclusion and exclusion criteria for studies in this review are given in Box 1.

Box 1: Inclusion and exclusion criteria of studies

Inclusion criteria

- Studies with inpatients, outpatients, and the general public about their perspectives/ opinions/views on the financial/material relationship between physicians and pharmaceutical companies.

- Empirical studies employing quantitative and qualitative methods.

- Journal articles published between 2003 and 2023.

- Studies only published in the English language.

Exclusion criteria

- Books, book chapters, editorials, opinions, and review articles.

- Grey literature such as information briefs, organizational reports, and theses.

- Studies seeking to understand the opinions of regulators and healthcare professionals such as physicians and pharmaceutical professionals.

- Studies published before 2003 to specifically look for published work within the last 20 years.

Stage 1 involved designing the search strategy and crafting research questions. Based on our preliminary search through Google Scholar, we found that a limited number of studies on this issue were available. Thus, our main question was what existing studies say about how HCCs perceived pharmaceutical incentivisation to physicians. As we performed a preliminary scan

of the literature, we noticed many examples of HCCs' perceptions of pharmaceutical incentivisation to physicians, leading to our next question of how HCCs understand the incentivisation practice with respect to different countries. It is usual for researchers to propose potential actions for future research, policy, and practice. However, we were interested in exploring what ways HCCs think of addressing ILP.

At stage 2, we began our formal search of the following databases: Scopus, Medline (OVID) and EMBASE (OVID). This search was conducted by an experienced academic health sciences liaison librarian [JL] on September 12, 2023, using a combination of keywords and subject headings (see S1 File). Limits were applied to retrieve results of citations to articles published in English from 2003 to the date of the search. The earliest study matching our criteria was in 2006. Additional searching was carried out via Google Scholar (first 10 pages) [by MNN] examining documents related to the key publications identified in the database searches, but we were unable to find any new studies.

At stage 3, from each database, we generated RIS files containing citations that were imported into Covidence–a website-based software to manage and carry out research reviews [18]. To identify the relevant studies, search results from Scopus, Medline (OVID) and EMBASE (OVID) were imported into Covidence [18]. Then, duplicates were identified and removed first using the automated tool in Covidence, and then manually by the research team. Title and abstract screening of the remaining studies was performed by a research specialist (HSA), to remove ineligible studies, following which, a full-text screening was performed by two team members (MNN and HSA) collaboratively, to determine the final number of eligible studies for this review. The study selection process is illustrated in Fig 1.

At stage 4, the data extraction and charting were performed on a Microsoft Excel Spreadsheet by three team members (HSA, ZA, and NVM). The team lead (MNN) then reviewed the data extraction matrix and resolved contradictions. The data matrix is given as a S2 File. We first screened for and recorded basic study information (year, setting, method, sample, and key findings) and HCCs' characteristics (number of participants, and distribution of participants with respect to age and gender). Here, we also segregated the eligible studies to the research designs used to conduct them. In 11 studies a descriptive research design was used, of which 10 were cross-sectional survey studies, and 1 was a qualitative study) [10,19–28]. In 2 studies, an analytical research design was used, of which 1 was experimental and 1 was an observational cohort study [29,30].

Finally, at stage 5, in line with the research questions, the team lead (MNN) conceptualised and reorganised various themes presented in the data matrix. The final condensed version of the data matrix guided the reporting of the results.

## Results

### Overview of studies

We identified a total of 564 references from databases such as Scopus, Medline (OVID) and EMBASE (OVID), of which 78 duplicates were removed. After the title and abstract screening of the remaining 486 studies, we removed 461 studies that did not meet the eligibility criteria. A full-text screening of the remaining 25 studies was performed by two team members (MNN and HSA), from which we further excluded 13 articles. Of these 13 excluded studies, 10 studies were conducted with population groups other than patients and the general public, 2 studies were non-empirical, and 1 study was not focused on perceptions of ILP.

The eligible studies were published between 2006 and 2022, from the USA (7), Australia (1), Lebanon (1), Malaysia (1), Pakistan (1), South Africa (1) and Turkey (1). In 12 studies, quantitative methods were used, while only 1 study had a qualitative research design. Ten

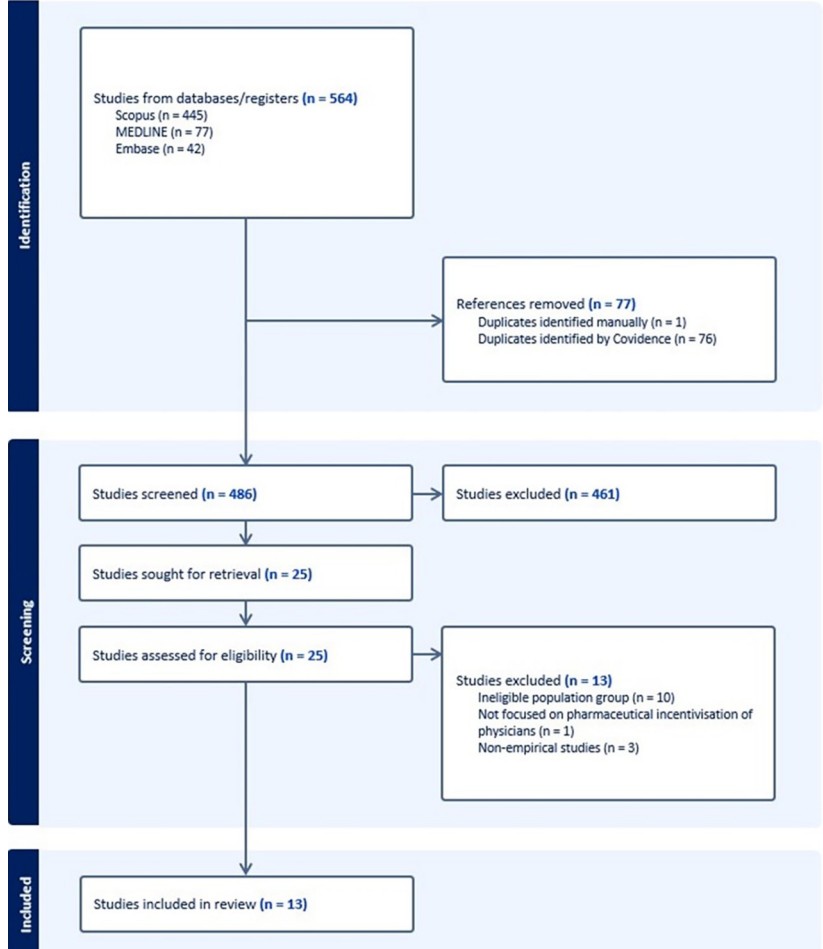

**Fig 1. Flowchart of the study selection process.**

studies were conducted with patients, while 4 studies were carried out with the general public [10,21,23,29]. The sample size in quantitative studies ranged from 192 to 3852 participants, whereas the qualitative study was conducted with 50 participants. In quantitative studies, a total of 4965 (51%) females and 4802 (49%) males participated, whereas the qualitative study was conducted with 28 female and 22 male participants. The age of the participants across all studies ranged from 18–70 years old. A detailed description of the characteristics of the eligible studies is given in Table 1.

Guided by our research questions, we explored three major themes identified in the review of eligible studies. The first theme was about HCCs' understanding of pharmaceutical incentivisation to physicians. Here, we organised information regarding the extent to which HCCs were aware of the types of incentives that pharmaceutical companies typically provided to physicians, the mechanism through which incentivisation happened, and factors associated with the level of HCCs' awareness about pharmaceutical incentivisation to physicians. The second theme contains information about HCCs' perceptions of the risks associated with physicians' engagement in ILP. The third theme synthesised information about HCCs' proposed actions for policy and regulation required to address pharmaceutical incentivisation to physicians. An outline of these interrelated themes is given in Fig 2.

**Table 1. Characteristics of the eligible studies.**

| Study | Approach | Aims | Country | Population | Sample | Key findings |
|---|---|---|---|---|---|---|
| Semin et al. (2006) [20] | Quantitative | To investigate the patients' opinions on the promotional activities of pharmaceutical companies. | Turkey | Patients | 584 | 1. Patients had a great awareness of pharmaceutical incentivisation to physicians. 2. Patients thought pharmaceutical incentivisation to physicians is an unethical practice. 3. Patients believed pharmaceutical incentivisation can affect physicians' prescribing behaviour. |
| Goff et al. (2008) [25] | Qualitative | To explore patients' beliefs and preferences about medication prescribing to understand factors that might affect medication adherence. | USA | Patients | 50 | 1. Patients believed that pharmaceutical companies have a great influence on physicians. |
| Tattersall et al. (2009) [19] | Quantitative | To seek the views of patients attending general practice about doctors' interactions with the pharmaceutical industry and their wishes for disclosure of this information. | Australia | Patients | 906 | 1. Patients had a patchy knowledge about incentive-linked prescribing. 2. Patients wanted to know if their doctor obtained any benefits in cash or kind from the pharmaceutical industry. 3. Physicians' disclosure of competing interests is important to help patients make informed treatment decisions. |
| Jastifer et al. (2009) [28] | Quantitative | To examine the general public's attitudes toward and awareness of physicians' acceptance of gifts from the pharmaceutical industry. | USA | Patients | 903 | 1. Participants reported various incentive types given to physicians from pharmaceutical companies such as drug samples, ballpoint pens, books, meals, and sponsorships for travel. 2. A majority of participants disapproved of gifts of a higher value such as travel sponsorships. |
| Crigger et al. (2009) [23] | Quantitative | To explore public perceptions of health care providers' role in pharmaceutical marketing. | USA | General Public | 223 | 1. Participants believed that their healthcare providers' prescribing practices were influenced by pharmaceutical representatives. 2. Participants were supportive of gifts for educational purposes. |
| Grande et al. (2012) [26] | Quantitative | To measure patient perceptions about the prevalence of industry gifts and their relationship to trust in doctors and the health care system. | USA | Patients | 2029 | 1. 34% of the participants believe almost all doctors receive gifts. 2. Participants of higher socioeconomic status (income, education) and younger age were more likely to believe their physician receives gifts.3. Participants who believed their physician received gifts were more likely to report low trust in physicians. |
| Green et al. (2012) [27] | Quantitative | To explore patients' awareness of interactions between physicians and the pharmaceutical industry and to examine whether those interactions impact trust and the doctor-patient relationship. | USA | Patients | 192 | 1. A majority of patients were unaware of the financial relationship between pharmaceutical companies and physicians. 2. Patients reported having less trust in physicians who accepted gifts from pharmaceutical companies. 3. Some patients reoprted that they would be less likely to take a prescribed medication if their physician had recently accepted gifts. |
| Wise et al. (2013) [30] | Quantitative | To examine patients' perceptions of the practice of physicians accepting gifts from the pharmaceutical industry. | South Africa | Patients | 200 | 1. Patients felt that it was unacceptable for physicians to accept a gift from a pharmaceutical Company. 2. A majority of patients believed that doctors were influenced by accepting gifts. 3. A majority of patients preferred to be cared for by a doctor who had no relationship with or did not accept gifts from, pharmaceutical companies. |

(*Continued*)

**Table 1.** (Continued)

| Study | Approach | Aims | Country | Population | Sample | Key findings |
|---|---|---|---|---|---|---|
| Perry et al. (2014) [22] | Mixed methods | To explore how patients perceive payments made by drug and device companies to physicians. | USA | Patients | 881 | 1. Payments had a significant effect on patients' trust in physicians. 2. Patients were less likely to identify ethical conflict if they perceived themselves as potential beneficiaries of the free drug samples. |
| Ammous et al. (2017) [10] | Quantitative | To assess the awareness and attitudes of the general public in Lebanon regarding the interactions between physicians and pharmaceutical companies. | Lebanon | General Public | 263 | 1. A majority of patients were aware of pharmaceutical company presence in physicians' offices. 2. A smaller percentage of participants were aware of the gift-related practices of physicians. 3. Patients' level of trust was affected if physicians accepted gifts of a higher value from pharmaceutical companies. |
| Hwong et al. (2017) [29] | Quantitative | To assess how viewing online public disclosure of industry payments affects patients' trust ratings for physicians, the medical profession, and the pharmaceutical and medical device industry. | USA | General Public | 278 | 1. Physicians who received payments received lower ratings for honesty and fidelity as compared to physicians who received no payments. 2. The disclosure website did not affect trust ratings for the medical profession or industry. |
| Gillani et al. (2022) [22] | Quantitative | To explore patient perceptions and attitudes regarding physician–pharmaceutical company interactions. | Pakistan | Patients | 3852 | 1. A large number of patients were aware of physician–pharmaceutical company interactions. 2. A small number of participants were aware of the financial relationship between physicians and pharmaceutical companies. |
| Kaur et al. (2022) [21] | Quantitative | To explore the Malaysian public's perceptions towards these relationships between physicians and the medical manufacturing industry. | Malaysia | General Public | 361 | 1. More than half of the participants were aware of the relationships between physicians and the pharmaceutical industry. 2. Online platforms were believed to be the preferred ways for physicians' disclosure of financial ties with the pharmaceutical industry. |

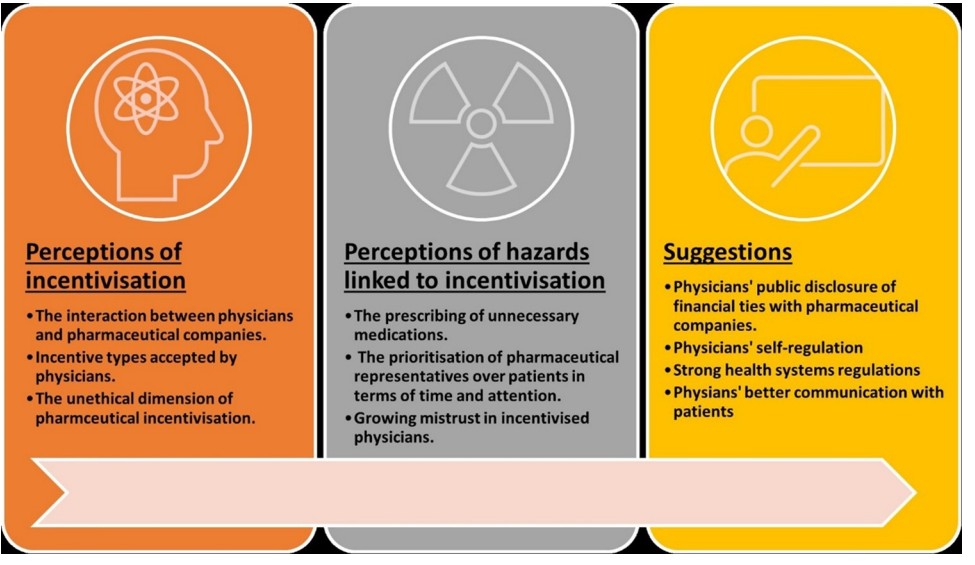

**Fig 2. Illustration of HCCs' perceptions of pharmaceutical incentivisation to physicians.**

### Theme 1: Understanding of pharmaceutical incentivisation

The level of HCCs' awareness about pharmaceutical incentivisation to physicians was assessed in 9 studies. Six studies conducted in the USA, Lebanon, Pakistan, Malaysia, and Turkey, showed a greater HCC awareness of pharmaceutical incentivisation to physicians [10,20–24,26]. For instance, in a study in the USA, 55% of the participants believed that their physicians receive gifts from pharmaceutical companies, whereas 34% thought all physicians do so [26]. Studies from Pakistan and Lebanon reported that more than half of the participants (Pakistan: 50.1%; Lebanon: 53%) witnessed items with pharmaceutical company logos on them in physicians' offices. One study in Australia, however, showed a low awareness among participants regarding pharmaceutical incentivisation. In this study, 76% of the participants were unaware of any financial relationship between physicians and pharmaceutical companies, and 84% of them believed that physicians' disclosure of their financial engagement with pharmaceutical companies would help patients make informed treatment decisions [19].

The types of incentives that pharmaceutical companies typically provide to physicians were mentioned by HCCs in 5 studies [10,21–24]. For example, in Ammous et al's study in Lebanon, a majority of participants did not know whether their physicians take any incentives/benefits from pharmaceutical companies, although 44% of them believed gifts might influence their physicians' prescribing behaviour [10]. In Pakistan, HCCs believed physicians take a range of incentives/benefits from pharmaceutical companies such as items valued at USD 90 or more (16.8%), participation in social activities organised by pharmaceutical companies (30.1%), paid trips (13.3%), meals (20%), and notepads/pens (49.9%) [24]. In Malaysia, HCCs pointed to free drug samples, educational materials, and the ownership of company stocks as other pharmaceutical incentives given to physicians [21]. In two US-based studies, HCCs spoke about the incentives they believed were acceptable or unacceptable for physicians to take from pharmaceutical companies [26,28]. Most of the HCCs in these studies approved of incentives of lesser value such as ball-point pens and free drug samples. The acceptability of these items was attributed to the idea that HCCs might view pens as trivial and unable to influence physicians' prescribing behaviour, whereas free drug samples might be viewed as a benefit to patients. Also, in Pakistan, lecturing or researching for pharmaceutical companies in exchange for money was considered acceptable, for over half of the participants (53.3%) [24].

In most of the studies, HCCs doubted the professionalism of physicians who accepted incentives from pharmaceutical companies, as this practice was perceived against the professional ethics in medical practice [10,20,21,23,24,27,30]. In the US, HCCs perceived physicians who had financial ties with pharmaceutical companies as dishonest, and hence not working in the best interests of HCCs [29].

### Theme 2: Perceptions of hazards linked to incentivisation

The perception of the influence of pharmaceutical incentivisation on physicians' prescribing behaviour was noted in nine studies [10,19–21,23,24,27,28,30]. Even gifts of lesser value were believed to distort physicians' prescribing behaviour in Pakistan and Lebanon [10,24], leading patients to consider prescriptions less reliable [20]. Therefore, in the case where a smaller number of HCCs were aware, they wanted to know if their physicians had financial ties with pharmaceutical companies [19]. Nevertheless, in some studies, a smaller number of HCCs reported a lower trust in their physicians, even though they thought incentivisation could influence their prescribing behaviours [10,21,24]. One of the reasons explained in those studies was that HCCs might find the relationship between physicians and pharmaceutical companies from the perspective of knowledge exchange [10,24]. However, differences in the level of trust were noted with respect to HCCs' level of knowledge, race, and ethnicity [19,21]. Although

physicians' disclosure of receiving payments from pharmaceutical companies might influence perceptions of honesty and professionalism, higher payments negatively affected their trust in physicians [22,29,30]. In some studies, HCCs believed that physicians who receive incentives from pharmaceutical companies could prioritise pharmaceutical company representatives over HCCs. This practice could further add to HCCs' problems, as they experience a longer waiting time while physicians interact with pharmaceutical company representatives [10,20,21,23–25,27].

### Theme 3: HCCs' suggestions for policy and practice

HCCs demanded transparency regarding physicians' financial ties with pharmaceutical companies, so they could maintain their trust in physicians [19,21,25]. HCCs also put a strong emphasis on better regulatory controls, so the relationship between physicians and pharmaceutical companies became transparent, which could increase their trust in healthcare systems [10,20,21]. The suggestions about public awareness campaigns regarding pharmaceutical incentivisation were also noted in a few studies, as HCCs believed better communication with physicians was contingent on better awareness about healthcare systems [10,21]. Furthermore, HCCs thought physicians needed to work in the best interests of HCCs and ensure to self-regulate their interactions with pharmaceutical companies [10,30]. In one study, HCCs believed that it was highly necessary for physicians to clearly communicate information about the medications they prescribe because doing so could create a sense of professionalism, empathy, and trust [25].

## Discussion

In this review, we grouped HCCs' perceptions into three broader categories: understanding of pharmaceutical incentivisation of physicians, perceptions of risks linked to ILP, and suggestions to improve the transparency of physicians. While many studies spoke about various incentives that physicians may take from pharmaceutical companies, patients in two US-based studies distinguished between acceptable/unacceptable incentives [26,28]. In these studies, items of a lesser value were believed to be acceptable, whereas things such as money, meals, and sponsorships of travelling were considered unacceptable. Although opinions vary when it comes to differentiating between what is ethical to take and what is not, it is generally considered acceptable for physicians to receive any items of minimal value from drug companies [31]. In consonance with this, patients in many eligible studies thought that it was acceptable for physicians to receive things like free drug samples, pens, and notepads. Consistent with the US Federal Anti-Kickbacks Statute (AKS) [32], patients in some studies believed it is unacceptable for physicians to receive cash and/or expensive items from pharmaceutical companies. The AKS is a criminal law, according to which incentives of a higher value such as money, free rent, expensive hotel stays and meals are illegal to take from drug manufacturers and suppliers [32].

HCCs' awareness of pharmaceutical incentivisation to physicians leads to several perceptions of hazards linked to it such as distorted prescribing behaviour, growing mistrust in physicians, and a lack of attention paid to patients. In many studies, HCCs believed that when physicians were incentivised, they would aim to meet pharmaceutical sales targets, even if they would need to prescribe medications unnecessarily [10,21,23,24,28]. The sense of incentive-driven prescriptions can further produce contexts for HCCs to reduce their trust in physicians. There is strong evidence that HCCs who lose trust in physicians are less likely to adhere to medical treatments recommended to them–something that may place a significant morbidity and financial burden on healthcare systems [33–35]. The presence of pharmaceutical sales

representatives and promotional items in physicians' offices, while HCCs wait to see their physicians, as explained in Ammous et al.'s [10] study, is one of the ways that may contribute to HCCs' awareness of pharmaceutical incentivisation. Hence, trust is a basic element of a healthy and respectful relationship between physicians and patients. Even if patients are reasonably health literate, they rely on physicians who are experts in the field of medicine [36]. The reduction in HCCs' trust is a clear sign of how the acceptance of pharmaceutical incentives can affect physicians' reputations in society. Therefore, HCCs in many studies expressed their interest in knowing whether their physicians had any financial ties with pharmaceutical companies, as the non-disclosure might be taken as an injustice.

We also found that HCCs in most of the studies were reasonably aware of the mechanisms through which pharmaceutical incentivisation takes place and the types of incentives that physicians typically receive from pharmaceutical companies. This finding is consistent with several empirical studies with physicians, pharmaceutical companies, and other relevant stakeholders who testify to the unethical exchanges between physicians and pharmaceutical companies in various parts of the world [4,8,13,37,38]. This means that even though physicians and pharmaceutical companies may attempt to establish financial ties discretely, HCCs have a reasonable awareness of this practice.

The acceptance of incentives from pharmaceutical companies was also deemed against professional medical ethics. This has implications for not only developing clear-cut guidelines on the ethics of dealing with pharmaceutical companies but also ensuring that physicians can fully understand and follow them. In many LMICs, such as Pakistan, no clear guidelines on professional medical ethics exist, and physicians often find it difficult to determine ethical boundaries when they interact with pharmaceutical sales representatives who visit them to promote drugs [7]. Therefore, the development of guidelines on ethics should be the prime responsibility of states to help physicians recognise potential conflicts of interest while dealing with physicians.

A few studies also showed that HCCs maintained their trust if their physicians accepted incentives of lesser value or small payments in exchange for lecturing or researching for pharmaceutical companies. This means that HCCs are aware of the importance of the interaction between the pharmaceutical industry and physicians, which happens around knowledge exchange and scientific development.

We limited our search to English-language scholarly journal articles indexed in the databases like Scopus, Medline (OVID), EMBASE (OVID), and a search of Google Scholar. The extension of the search to include research in other languages, and the inclusion of grey literature could help to discover more information. Studies that we included in the review varied in terms of settings and methods, which we analysed to present a general landscape of the published literature on HCCs' perceptions of pharmaceutical incentivisation to physicians. A full systematic review would provide a more rigorous analysis of the studies.

Our review has several implications for future research. One of our major findings is that most of the empirical studies conducted with HCCs are underpinned by a positivist approach. Quantitative methods were useful in that they helped researchers to determine the distribution of the sampled HCCs about their beliefs and knowledge about pharmaceutical incentivisation and how their beliefs/knowledge might affect their attitudes toward physicians. However, qualitative research can prove more useful in explaining social conditions and contexts linked to HCCs' perceptions, which can further provide rich insights for policy and practice. More than half of the studies in our review came from high-income countries such as the USA and Australia. The problem of pharmaceutical incentivisation is more critical in LMICs due to weak regulatory controls. More future studies in LMICs are required for a better understanding of

the dynamics of pharmaceutical incentivisation to physicians, so appropriate interventions are developed to address this problem.

## Conclusion

HCCs in different parts of the world seem to be reasonably aware of the unethical financial relationship that is sometimes established between physicians and pharmaceutical companies. It is therefore necessary for physicians to avoid engaging in ILP. In the case where physicians' professional services are required in the pharmaceutical industry, they must publicly disclose this, so their patients maintain trust. The deterioration of trust due to physicians' engagement in ILP can not only reduce their adherence to treatments recommended by physicians but also negatively affect physicians' reputations in society. Finally, we emphasise the importance of health system research representing HCCs' voices to shape and strengthen policy, regulation, and practice.

## Supporting information

**S1 File. Literature search strategy.**
(DOCX)

**S2 File. Data matrix.**
(DOCX)

**S3 File. PRISMA ScR-checklist.**
(DOCX)

## Author Contributions

**Conceptualization:** Muhammad Naveed Noor, Haider Safdar Abbasi, Mishal Sameer Khan, Sadia Shakoor.

**Data curation:** Muhammad Naveed Noor, Haider Safdar Abbasi.

**Formal analysis:** Muhammad Naveed Noor, Nina van Der Mark, Zahida Azizullah.

**Funding acquisition:** Mishal Sameer Khan.

**Investigation:** Muhammad Naveed Noor.

**Methodology:** Muhammad Naveed Noor, Janice Linton.

**Project administration:** Muhammad Naveed Noor, Sadia Shakoor.

**Software:** Muhammad Naveed Noor, Haider Safdar Abbasi, Nina van Der Mark.

**Supervision:** Muhammad Naveed Noor.

**Visualization:** Muhammad Naveed Noor.

**Writing – original draft:** Muhammad Naveed Noor.

**Writing – review & editing:** Afifah Rahman-Shepherd, Amna Rehana Siddiqui, Mishal Sameer Khan, Rumina Hasan, Sadia Shakoor.

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
