## [Decision Letter · Decision Letter 0]

3 Apr 2024

PGPH-D-24-00380

Healthcare Consumers’ Perceptions of Incentive-Linked Prescribing: A Scoping Review of Research

Dear Dr. Khan,

Thank you for submitting your manuscript to PLOS Global Public Health. After careful consideration, we feel that it has merit but does not fully meet PLOS Global Public Health’s publication criteria as it currently stands. Therefore, we invite you to submit a revised version of the manuscript that addresses the points raised during the review process.

Please address the concerns raised by the reviewer, specifically:

- state the design of the reviewed studies

- in discussion, expand on both ethical and unethical promotional activities as well as perceived incentives within the framework of ethical and unethical practices (as defined by Federal anti-kickback laws).

We look forward to receiving your revised manuscript.

Kind regards,

Hassan Haghparast Bidgoli

Academic Editor

Journal Requirements:

Reviewers' comments:

Reviewer's Responses to Questions

**Comments to the Author**

1. Does this manuscript meet PLOS Global Public Health’s publication criteria? Is the manuscript technically sound, and do the data support the conclusions? The manuscript must describe methodologically and ethically rigorous research with conclusions that are appropriately drawn based on the data presented.

Reviewer #1: Yes

Reviewer #2: Yes

2. Has the statistical analysis been performed appropriately and rigorously?

Reviewer #1: N/A

Reviewer #2: N/A

3. Have the authors made all data underlying the findings in their manuscript fully available (please refer to the Data Availability Statement at the start of the manuscript PDF file)?

Reviewer #1: Yes

Reviewer #2: Yes

4. Is the manuscript presented in an intelligible fashion and written in standard English?

Reviewer #1: Yes

Reviewer #2: Yes

5. Review Comments to the Author

Reviewer #1: The title and the objective highlighted in this scoping review is important to understand pharmaceutical promotion from the patient and general public perspective. However, I recommend some modifications in order to enrich the work. First, use a uniform referencing style or put the reference citation before the full stop. Second, reduce the unnecessary use of pronouns and present perfect tenses. Furthermore, address the point listed below.

Introduction

#1: the idea stated from line 56 to 61 is too basic for a research article.

Methods

#2: why did you only include articles publish from 2003 onwards?

#3: If possible, remove box 2 and attach it as additional files.

#4: why didn’t you scrutinize articles by study design in stage 4?

Discussion

#5: Discuss the ethical promotional vs unethical promotional activities. Otherwise, some of the activities highlighted by this study could not be considered as incentivizing the physicians. For example, Providing product sample is ethical promotional activity and as long as the physician did not use the sample for treatment purpose is not an incentive.

Reviewer #2: The authors have done justice to this scoping review which interogatted the knowledge and views of healthcare consumers (patients) about the interaction between physicians and the pharmarceutical industry.

The manuscript followed the steps recommended for a scoping review and the deductions were spot on.

6. PLOS authors have the option to publish the peer review history of their article (what does this mean?). If published, this will include your full peer review and any attached files.

**Do you want your identity to be public for this peer review?** For information about this choice, including consent withdrawal, please see our Privacy Policy.

Reviewer #1: **Yes: **Dr. Bereket Molla Tigabu

Reviewer #2: **Yes: **Prof. Joseph O. Fadare

---

## [Decision Letter · Decision Letter 1]

21 May 2024

Healthcare Consumers’ Perceptions of Incentive-Linked Prescribing: A Scoping Review

PGPH-D-24-00380R1

Dear Dr. Khan,

We are pleased to inform you that your manuscript 'Healthcare Consumers’ Perceptions of Incentive-Linked Prescribing: A Scoping Review' has been provisionally accepted for publication in PLOS Global Public Health.

Best regards,

Hassan Haghparast Bidgoli

Academic Editor

Reviewer Comments (if any, and for reference):

Reviewer's Responses to Questions

**Comments to the Author**

1. If the authors have adequately addressed your comments raised in a previous round of review and you feel that this manuscript is now acceptable for publication, you may indicate that here to bypass the “Comments to the Author” section, enter your conflict of interest statement in the “Confidential to Editor” section, and submit your "Accept" recommendation.

Reviewer #1: All comments have been addressed

2. Does this manuscript meet PLOS Global Public Health’s publication criteria? Is the manuscript technically sound, and do the data support the conclusions? The manuscript must describe methodologically and ethically rigorous research with conclusions that are appropriately drawn based on the data presented.

Reviewer #1: Yes

3. Has the statistical analysis been performed appropriately and rigorously?

Reviewer #1: N/A

4. Have the authors made all data underlying the findings in their manuscript fully available (please refer to the Data Availability Statement at the start of the manuscript PDF file)?

Reviewer #1: Yes

5. Is the manuscript presented in an intelligible fashion and written in standard English?

Reviewer #1: (No Response)

6. Review Comments to the Author

Reviewer #1: No more comments

7. PLOS authors have the option to publish the peer review history of their article (what does this mean?). If published, this will include your full peer review and any attached files.

**Do you want your identity to be public for this peer review?** For information about this choice, including consent withdrawal, please see our Privacy Policy.

Reviewer #1: No
